# Learned Prefix Caching for Efficient LLM Inference

**Dongsheng Yang**   **Austin Li**   **Kai Li**   **Wyatt Lloyd**
Department of Computer Science
Princeton University
{dongsheng, al2926, li, wlloyd}@princeton.edu

## Abstract

Prefix caching is a key technique for reducing Large Language Model (LLM) inference costs. However, the prevalent least-recently-used (LRU) eviction algorithm has a large gap to the optimal algorithm. This paper introduces LPC, the first learned method to perform LLM prefix cache eviction. LPC leverages conversational content analysis to provide predictive guidance for eviction, determining which conversations are likely to continue. These insights, combined with last access timestamps, inform more effective cache management. Extensive evaluations across three real-world datasets demonstrate that LPC achieves 18–47% reductions in required cache sizes for equivalent hit ratios and has an 11% improvement in LLM prefilling throughput in an emulated environment.

## 1 Introduction

Prefix caching has been recently proposed as an optimization for reducing inference latency and compute cost in Large Language Model (LLM) serving systems [Gim et al., 2024, Gao et al., 2024]. By storing and reusing internal Key-Value (KV) states from previously processed input prefixes, prefix caches can avoid redundant computation during the expensive prefilling phase of inference, especially in multi-turn conversations where historical context is repeatedly reused. A cache hit on a long context can reduce Time-To-First-Token (TTFT) latency by 74% as shown in Section 4.5. In a production environment serving millions of requests, high hit ratios can significantly increase server throughput, translating to substantial hardware cost savings and accommodating thousands of additional users on the same infrastructure.

Despite these benefits, the effectiveness of current prefix caching systems [Kwon et al., 2023, Zheng et al., 2024] is constrained by their least-recently-used (LRU) eviction strategy, which discards the least recently accessed KV blocks. Although simple and widely adopted, LRU's performance is far from optimal for LLM prefix caching. As illustrated in Figure 1, there is a large gap between the hit ratios achieved by LRU and an idealized Oracle policy that possesses perfect knowledge of future conversation continuations.

The success of learned caching policies in other domains, such as Content Delivery Networks (CDNs), suggests that learned cache designs can significantly outperform LRU and other heuristics on hit ratios [Song et al., 2020, Yang et al., 2023, Song et al., 2023]. However, the fundamentally different nature of LLM workloads prevents a direct transplantation of these techniques. LLM prefix caches manage semantically interconnected, variable-length segments of conversational context, which are far more complex than the relatively independent, often fixed-size objects handled by traditional learned caches in

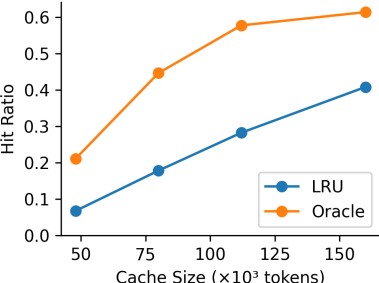

Figure 1: A substantial gap between the hit ratios achieved by LRU and an idealized Oracle policy with perfect knowledge of future conversation continuation (LMSys dataset).

39th Conference on Neural Information Processing Systems (NeurIPS 2025).

other domains. This disparity necessitates the development of novel, specialized learned approaches tailored to the unique characteristics of LLM workloads.

To bridge this gap, we introduce LPC, a lightweight learned prefix caching framework specifically designed for LLM conversational systems. LPC addresses two primary challenges inherent in this domain. The first is to achieve substantially higher cache hit ratios over LRU-based approaches, by accurately predicting the likelihood of conversation continuation. The second is ensuring the learning component incurs negligible memory and computational overhead—a critical requirement for practical deployment within resource-constrained LLM serving environments—in order to deliver an overall lower inference latency and higher inference throughput.

Our contributions are as follows: (1) We propose the first learned prefix cache to leverage the opportunity to improve LLM inference efficiency by increasing prefix reuse. (2) We design a method to learn and predict conversation continuation probability with negligible memory and computational overhead. (3) Our evaluations of an implemented prototype system shows that LPC consistently outperforms LRU, achieving 18–47% reductions in the required cache size to achieve the same hit ratio. In an emulated disaggregated serving [Zhong et al., 2024] and reasoning [Snell et al., 2024] environment, this translates to LPC delivering up to 11% higher prefilling throughput and reducing the first token latency of up to 7% of requests by 42–75%. Furthermore, as a system-level cache management policy, LPC is orthogonal to and can be combined with other optimizations like FlashAttention or speculative decoding. The implementation can be found at `https://github.com/yangdsh/LPC`.

## 2 Background

LLMs employ caching mechanisms to store internal representations of the input sequence, known as the KV cache and the prefix cache. This section describes the roles of the KV cache and the prefix cache in LLM inference, and presents the challenges in prefix cache eviction.

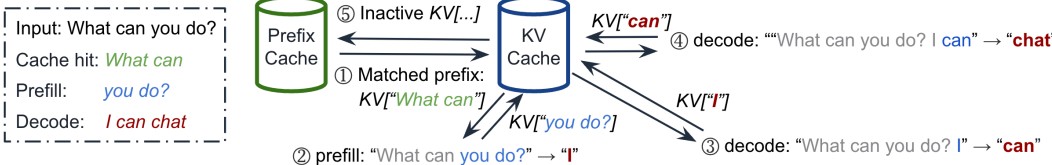

Figure 2: An example of the use of the KV cache and the prefix cache in LLM inference.

**KV cache for LLM inference.**    The KV cache stores Key-Value (KV) states to support autoregressive LLM inference. These KV states are computed representations of the contextual information of each token. The KV cache is heavily used during the two phases of LLM inference. Initially, the *prefilling* phase processes the entire input prompt. For each token in this prompt, say $token_i$, its corresponding KV states, $kv_i$, are computed and stored in the KV cache. This phase concludes with the generation of the first output token. Subsequently, the *decoding* phase utilizes KV states in the KV cache for all previous tokens of this sequence, representing the full context seen so far, to generate the next token. For efficient memory management and to reduce fragmentation, the KV cache often organizes these states into fixed-size "KV blocks", where the block size is typically around 16 tokens [Kwon et al., 2023]. Upon completion of a sequence, the KV states associated with that sequence are typically cleared from the KV cache. The primary role of this KV cache is thus to eliminate redundant computations for token contexts within an individual generation sequence.

**Prefix caching.**    The standard KV cache typically discards KV states upon request completion. This prevents reuse across related requests, such as requests continuing a previous conversation, forcing costly re-prefilling of shared contexts. The *prefix cache* addresses this by retaining KV states from completed sequences. When a new request arrives, the system queries the prefix cache for the longest matching prefix, for example $token_1, \ldots, token_m$, of the input. If its KV states $(kv_1, \ldots, kv_m)$ are found, they are made available to the KV cache for the current request. Prefilling is then only required for the remaining, unmatched portion of the input, from $token_{m+1}$ onwards, significantly reducing computation. An example is shown in Figure 2 where the prefilling of "What can" is skipped because it matches in the prefix cache.

**Prefix cache eviction.** The prefix cache has limited capacity as it uses precious GPU memory. When the prefix cache becomes full or the KV cache demands more memory, the system must evict existing KV blocks. The choice of which blocks to evict is crucial: a well-designed eviction policy directly improves the prefix caches *hit ratio*the fraction of future requests that find a matching prefixthereby reducing redundant computation and improving throughput and latency.

However, existing systems use the LRU policy, which evicts the least-recently used blocks. This approach, while reasonably effective in traditional caching domains, falls short in LLM serving environments. Prefix cache workloads exhibit different temporal and spatial locality patterns. In human-LLM interactions, users naturally pause between turns to read, think, or compose follow-up queries, meaning useful prefixes may go untouched for extended periods despite being highly relevant. Furthermore, LRU fails to capture the semantic or task-related context of a conversationfactors that are often more predictive of future reuse than recency alone. A conversation centered around a complex, unresolved task is more likely to continue than one involving a brief, close-ended exchange. These differences motivate us to design new eviction strategies that go beyond recency and account for the unique behavioral and semantic patterns in LLM workloads.

## 3 Learned Prefix Caching

This section presents a learned prefix caching approach to significantly advance the state-of-the-art in cache management for LLM inference. Our goal is to improve prefix cache effectiveness by intelligently selecting which Key-Value (KV) blocks to evict when space is needed, thereby reducing prefilling latency and increasing throughput. To achieve this, our method must satisfy three key requirements:

- **High Hit Ratios:** The approach should consistently and substantially outperform the LRU baseline in terms of cache hit rate across diverse workloads.

- **Improved Latency and Throughput:** The method must be efficient without introducing bottlenecks, enabling both lower latency and higher throughput.

- **Minimal Overhead:** The system should impose negligible overhead in terms of GPU memory usage and computational cost.

### 3.1 Overview of the LPC Framework

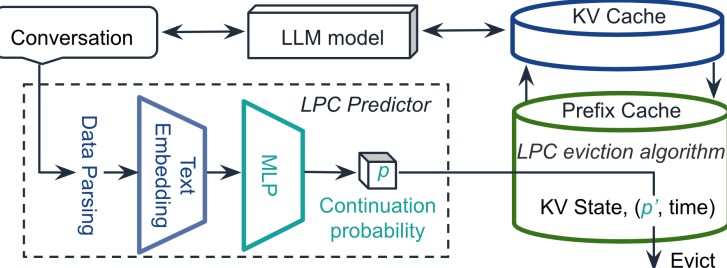

Figure 3: **Overview of LPC**. The LPC framework contains a predictor and an eviction algorithm.

We propose LPC, a learned prefix caching to meet the requirements above. Figure 3 illustrates the design of LPC. The main difference between this approach and a common prefix cache is that LPC uses a predictor to inform the replacement algorithm about which blocks to evict, instead of picking the least recently used ones.

The **LPC predictor** takes the conversation history and outputs a probability score indicating the likelihood of the conversation continuing. Its steps are:

1. Data parsing: extract the user prompts from the conversation history.

2. Text embedding: generate a semantic embedding vector from the parsed user prompts.

3. MLP-based classification: feed the text embedding and other features into a Multilayer Perceptron (MLP) classifier to predict the conversation continuation probability $p$.

The **LPC eviction algorithm** has two running components:

1. Cache Insertion and Eviction: insert KV blocks into a min-heap based on their updated probabilities $p'$; evict the KV block with the smallest $p'$ when the prefix cache is full.

2. Probability Update: periodically apply a decay function to the predicted continuation probability $p$ to get an updated probability $p'$ that factors in the elapsed time.

### 3.2 LPC Predictor

This section describes the three components of the LPC predictor: data parsing, the text embedding model, and the MLP classifier.

**Data parsing.**  To provide the model with sufficient context for accurate predictions, text and numerical features are extracted from the conversation. These include the text of the new user prompt and the user prompts of the preceding $N$ conversational turns. Our implementation uses $N = 4$ to keep much of contextual information available while keeping resource usage low. For similar reasons we do not include model responses because they contain less new information than user prompts, a finding validated by our ablation study in the appendix. Additionally, the conversation length, represented as the number of turns already occurred, is included as a scalar feature, which is especially informative for conversations with more than $N$ turns.

**Text embedding model.**  For transforming raw text into meaningful vector representations, we utilize the `multilingual-e5-small` model [Wang et al., 2024] (with MIT license) from the family of sentence transformer models. We choose this model because it is capable of representing important patterns but is highly efficient. This model has 118M parameters. Such efficient models often have constraints on input length; for `multilingual-e5-small`, this is 512 tokens. A key challenge, therefore, is to compress the input exceeding the length limit while preserving essential information. We employ a strategic token allocation as follows: the 512-token budget is distributed evenly across each of the $N + 1$ user request turns. For each turn's allocated token budget, half the tokens are sourced from the beginning of the prompt, and the other half from the end. This strategy is informed by research indicating that the most important information in textual input often lies in these extremities [Liu et al., 2023]. The output of the text embedding model is a vector with 384 dimensions.

**MLP-based classification Model.**  The classification model is a 3-layer MLP, each layer with 128 hidden neurons. We select it because it is very efficient while having sufficient capability to learn for our task. It accepts a concatenated feature vector as input, which consists of the 384-dimensional text embedding vector and a scalar value representing the number of conversation turns. The model outputs a single probability value, ranging from 0 to 1, indicating the confidence that the conversation will continue beyond the current turn.

We have made several design decisions to minimize the overhead of the predictor, First, we use small text embedding and MLP models that have low GPU memory overhead. Second, the predictor runs only once for each user request, where the request rate is typically lower than 10 per second per GPU. Third, the predictor is a stand-alone component **running in parallel** to LLM inference. Lastly, one predictor can pair with multiple LLM inference instances to further amortize the memory overhead and computational overhead through batching.

### 3.3 Training

This section outlines how LPC is trained, including training data collection, training configurations, and retraining.

**Training data.**  Training and validation data are collected from a dedicated partition of conversations from the target dataset (as described in Section 4.2). This partition is strictly isolated and excluded from any datasets used for online evaluation. The size of the partition is half of the dataset. A separate model is trained for each dataset.

**Training configurations.** The training works as follows. First, during training, only the weights of the MLP classifier are updated; the pre-trained text embedding model is kept frozen. This partial fine-tuning method allows the MLP to adapt specifically to the continuation prediction task while ensuring the framework remains lightweight. Full fine-tuning would be operationally expensive for frequent retraining, whereas training only the tiny MLP head is extremely fast (typically less than 10 minutes), making daily adaptation feasible. Second, text embeddings for the training dataset are pre-computed and cached after the first epoch to significantly accelerate training. Subsequent epochs directly reuse these cached embeddings because the text embedding model is frozen, bypassing repeated computations and saving 90% of the training time. A binary cross-entropy loss function is employed. The loss is weighted with more weights for the minority class to handle class imbalance. The optimizer is Adam with a learning rate of $5 \times 10^{-4}$. The training runs until convergence (within 20 epochs in our evaluation), and the checkpoint with the lowest loss on the validation dataset is saved for running online inference.

**Retraining.** The predictor is trained offline on historical data. To adapt to gradual shifts in user behavior or evolving conversational dynamics and mitigate the risk of performance degradation from out-of-distribution (OOD) data, periodic offline retraining is the primary strategy. We propose daily retraining, a standard practice for ML-based cache policies to adapt to evolving query patterns [Song et al., 2023]. Training takes about 10 minutes, making this schedule affordable. For use cases with extremely high OOD risk, such as applications driven by rapidly changing current events, the system could leverage online learning to update itself incrementally. However, for typical conversational workloads, the general patterns of continuation are relatively stable, making frequent offline retraining a sufficient and operationally simpler choice.

### 3.4 Eviction Algorithm

This section introduces the LPC eviction algorithm in more detail. Our design largely follows the LRU prefix cache eviction algorithm but uses a dynamic conversation continuation probability instead of the last access time to rank objects.

#### 3.4.1 Data Structure and Operations.

The key data structure is a min-heap. Blocks in the heap are prioritized based on a dynamically calculated continuation probability of their associated conversation. The primary inputs associated with each sequence's KV blocks are its initial predicted continuation probability and its last access timestamp.

The operations are:

- **Insertion:** When the decoding of a sequence completes, its KV blocks are removed from the KV cache and inserted into the prefix cache's min-heap according to their initial predicted continuation probability.

- **Promotion:** If a prefix of an incoming user request matches a sequence of KV blocks in the prefix cache, these blocks are considered "hit." They are then promoted to the KV cache and are consequently removed from the prefix cache's min-heap.

- **Probability update:** The algorithm performs a periodic probability decay to be able to evict the KV blocks with over-estimated continuation probability. At configurable fixed intervals (defaulting to 10 seconds), LPC iterates through all KV blocks in the prefix cache. For each block, its associated continuation probability is re-evaluated using a decay function below. The min-heap is then reorganized to accurately reflect these updated probabilities.

- **Eviction:** When the prefix cache reaches its capacity, or when memory needs to be reclaimed (e.g., due to expansion of the KV cache), an eviction process is triggered. The KV block located at the root of the min-heap is selected and removed from the prefix cache.

#### 3.4.2 Probability Decay Function

To handle KV blocks with over-estimated continuation probability predictions from staying in the cache forever we periodically update their probabilities. We implement a decay function to adjust

the conversation continuation probability as time elapses since its last interaction. This function's input includes the initially predicted continuation probability ($prob_{\text{original}}$), the current system time ($time_{\text{cur}}$), and the timestamp of the last access to this KV block ($time_{\text{last}}$). Its output is the current, time-adjusted probability ($prob_{\text{cur}}$), which is calculated as:

$$prob_{\text{cur}} = \frac{prob_{\text{original}} \times decay}{prob_{\text{original}} \times decay + (1 - prob_{\text{original}})}$$

The *decay* factor itself is an exponential function of the elapsed time:

$$decay = \exp(-(time_{\text{cur}} - time_{\text{last}}) \times scale)$$

Here, *scale* is a hyperparameter controlling the decay rate. It was tuned empirically, and a good rule of thumb is to set it to be the inverse of the average turn interval. Since this interval is approximately 100 seconds across our datasets, we set $scale = 10^{-2}$ in all experiments. This mechanism ensures that prefixes from inactive conversations receive progressively lower continuation probabilities, making them more likely eviction candidates.

### 3.5 Handling KV Blocks Shared by Multiple Conversations.

A significant challenge in prefix caching arises with KV blocks shared across multiple distinct conversations, such as those resulting from common system prompts or identical initial user queries. Naive metadata management that simply overwrites the score of such blocks with the latest score can lead to suboptimal eviction decisions, because a popular prefix shared by multiple conversations would be incorrectly evicted by the most recent related conversation if it has a low score. Our algorithm incorporates specific logic for these shared blocks. The $time_{last}$ for a shared block is always updated to its most recent access timestamp. Furthermore, when a shared block is accessed as part of a new or existing conversation, its effective continuation probability for eviction decisions is set to the maximum of its $prob_{\text{cur}}^{\text{block}}$ calculated now from the previously stored metadata and $prob_{\text{cur}}^{\text{new}}$ calculated from the incoming metadata. This max-pooling operation across all conversations sharing the block is represented as:

$$prob_{\text{cur}}^{\text{block}} = \max(prob_{\text{cur}}^{\text{block}}, prob_{\text{cur}}^{\text{new}})$$

This strategy ensures that a block shared by multiple conversations, where at least one is highly likely to continue and has been recently active, is not prematurely evicted due to association with a less likely-to-continue conversation.

## 4 Evaluation

To evaluate how well LPC meets its design goals, we focus on three evaluation questions:

- How much does LPC improve prefix cache hit ratios compared to LRU? (4.4, 4.3)
- To what extent does the higher hit ratio of LPC translate into improved throughput and reduced latency? (4.5)
- What are the memory and computational overheads introduced by LPC? (4.6)

### 4.1 Implementation and Evaluation Environments

We implement a prototype of LPC on top of the vLLM serving framework [Kwon et al., 2023], which is licensed under Apache-2.0. The conversation continuation predictor runs as an external component, while the cache eviction policy is integrated directly into the vLLM prefix cache. Unless otherwise specified, all vLLM arguments use their default values. The implementation is based on the main branch of vLLM as of March 10, 2025.

We allocate 1 GB of GPU memory for the predictor used in LPC. This allocation breaks down as follows: approximately 560 MB for PyTorchs CUDA context and kernel initialization, 240 MB for loading the `multilingual-e5-small` model weights in float16 precision, and around 200 MB for runtime memory to store inputs and attention buffers.

All experiments were conducted on NVIDIA H100 GPUs, equipped with 80 GB of HBM3 memory. We use the `Qwen3-32B-FP8` model [Qwen3 Team, 2025] to run inference. It is a 32-billion

parameter reasoning model and is licensed with Apache 2.0. We use a single GPU to run the model. Considering the memory used by the model and pytorch, the memory available for KV cache and Prefix cache is 40 GB in total. The other hardware usage includes 8 CPUs and 64 GB CPU memory. The baseline is vLLM's LRU cache eviction algorithm.

Our evaluations use the following metrics:

- **Hit Ratio**: The average prefix cache hit ratio across all requests.
- **Time-to-First-Token (TTFT)**: The latency observed by the client from sending a request to receiving the first response token.
- **Throughput**: The average number of sequences prefilling per second.

To ensure a fair comparison with the LRU-based approach, **we deduct 1 GB from LPC's prefix cache size in all evaluations**, to account for its GPU memory overhead.

## 4.2 Workload

We run prefix cache hit ratio comparisons on real-world conversational datasets. These include:

- LMSys: a large-scale collection of user-chatbot dialogues [Zheng et al., 2023a].
- ShareGPT: conversations with ChatGPT [ShareGPT Team, 2023].
- Chatbot-Arena: a platform where users compare various LLMs [Chiang et al., 2024].

LMSys has a limited use license. ShareGPT has a Apache-2.0 license. Chatbot-Arena has a CC license. Table 1 summarizes the statistics of the three datasets. The follow-up ratio is the percentage of requests that will receive a future request of the same conversation.

Table 1: Dataset Properties.

| Dataset | Total conversations | Avg. input length | Avg. output length | Follow-up ratio |
|---|---|---|---|---|
| LMSys | $1 \times 10^6$ | 63 tokens | 179 tokens | 35% |
| ShareGPT | $9.5 \times 10^4$ | 113 tokens | 305 tokens | 55% |
| Chatbot-Arena | $3.3 \times 10^4$ | 36 tokens | 155 tokens | 8% |

An important limitation of public datasets is the lack of timestamp information, making it difficult to evaluate LLM systems under realistic conversational workloads. Simple Poisson-based arrival models are insufficient: they ignore LLM generation timespotentially placing a user request before the previous response finishesand do not account for conversation start times.

To overcome this limitation, we introduce a two-phase strategy that separates user think time modeling from LLM response times and schedules conversation start times to simulate realistic server load. In the pre-run phase, chat intervals within each conversation are sampled from an exponential distribution with mean $\lambda_{chat}$, and conversation start times are spaced to keep the number of concurrent active conversations below a target threshold $N_{conv}$ (default 200), emulating a static load balancer.

In the runtime phase, each conversation starts at its assigned time, and subsequent requests occur after adding the sampled user interval to the actual completion time of the previous LLM response. Evaluations run for 20 minutes to avoid capturing the cool-down period at the end of the trace.

## 4.3 Hit Ratios Across Varying Cache Sizes

To understand how LPC performs under different memory budgets, we compare its prefix cache hit ratio with LRU's while varying total token capacity. We vary the combined size of the prefix and KV caches from 60K to 160K tokens (approximately 15.6 GB to 40 GB). They are allocated across the prefix and KV caches dynamically by vLLM. The chat interval parameter $\lambda_{chat}$ is fixed at 100.

Figure 4 shows that LPC consistently outperforms LRU across all datasets and cache sizes. The improvement relative to LRU is 13% to 38% on LMSys (Figure 4a), 14% to 98% on ShareGPT (Figure 4b), and 15% to 30% on Chatbot-Arena (Figure 4c).

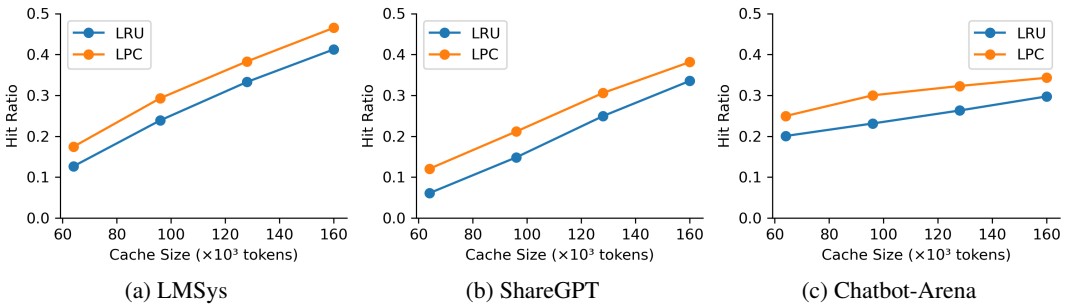

|     |     |     |
| --- | --- | --- |
| (a) LMSys | (b) ShareGPT | (c) Chatbot-Arena |

Figure 4: Hit ratio vs. cache size. LPC achieves consistently higher hit ratios than LRU, which translates to 18–47% reductions in the required cache size to achieve the same hit ratio as LRU.

To achieve the same hit ratio as LRU, LPC requires significantly less memory across all three datasets. For example, on Chatbot-Arena, LPC reaches a 30% hit ratio with only 85 K tokensjust 57% of the 150 K tokens required by LRU. Similar memory savings are observed on LMSys (up to 18% less) and ShareGPT (up to 30% less). This is a 7.2–18.8 GB GPU memory saving for a 40 GB cache. These results show LPC substantially improves memory efficiency in prefix caching.

## 4.4 Impact of Chat Intervals on Hit Ratio

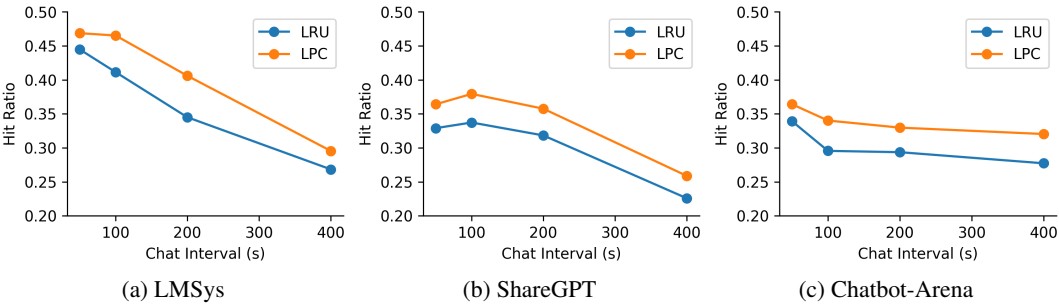

|     |     |     |
| --- | --- | --- |
| (a) LMSys | (b) ShareGPT | (c) Chatbot-Arena |

Figure 5: Hit ratio vs. chat intervals. LPC has consistently higher hit ratio than LRU, with 5–18% relative improvements.

To evaluate the robustness of LPC under varying user interaction patterns, we fix the cache size to 40 GB (the full available GPU memory) and vary the chat interval parameter $\lambda_{\text{chat}}$ to simulate different user think times. Shorter intervals represent more frequent user interactions, which may increase cache contention, while longer intervals simulate slower-paced conversations. with potentially higher reuse of cached prefixes.

As shown in Figure 5, LPC consistently outperforms LRU across all datasets and interval settings, achieving 5% to 18% higher hit ratios. Specifically, improvements range from 5% to 18% on LMSys (Figure 5a), 11% to 15% on ShareGPT (Figure 5b), and 7% to 16% on Chatbot-Arena (Figure 5c). These results show that LPC maintains strong performance across diverse conversational workloads, adapting well to varying user behaviors.

## 4.5 Throughput and Latency

To evaluate the impact on throughput and latency by the improved hit ratios, we use a microbenchmark that emulates a prefilling-only server in a disaggregated inference framework. This method is widely used in industry and is becoming a next-generation standard [Zhong et al., 2024, Patel et al., 2024, Hu et al., 2024, Qin et al., 2025]. A prefilling request rate under 10 req/s is a reasonable and often conservative estimate for production models with moderate prompt lengths [Zhong et al., 2024].

We use a synthetic workload with longer context lengths to reflect the recent trend of test-time scaling related to the wide adoption of reasoning models [Snell et al., 2024, Chen et al., 2025]. Recent

studies show that the generation length significantly increases from around 10 tokens generated by non-reasoning models to around 1,000 tokens generated by reasoning models [Han et al., 2024, Sun et al., 2025]. This increase in the generation lengths means an increase in the lengths of the matched contexts, because generated text is part of the context of the input of the next conversation round.

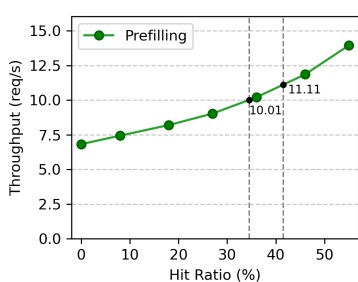

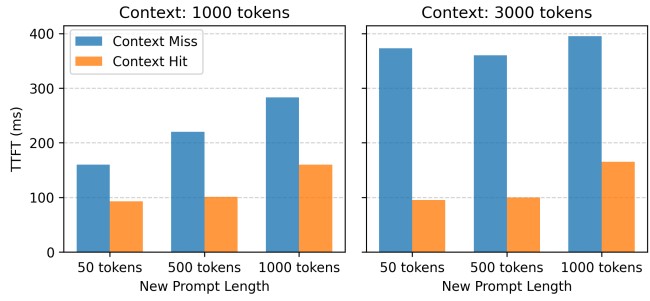

Figure 6: Prefilling throughput (req/s) as a function of hit ratio.

Figure 7: Time To First Token (TTFT) comparison for context cache miss vs. hit across varying context and prompt lengths.

Figure 6 shows the prefilling throughput gain as the hit ratio improves. In this microbenchmark, every input has a 1000-tokens context and a 50-token new prompt. The x-axis is the prefix cache hit ratio of the full contexts, including the new prompts that always miss. A hit ratio boost from 35 percent to 42 percent, which LPC achieves in Figure 5a, leads to an increase in throughput from 10.01 req/s to 11.11 req/s, indicating an 11.10% higher prefilling throughput for LPC over LRU in the emulated environment.

Figure 7 quantifies the reduction in TTFT achieved by a context cache hit. It shows that a context hit can keep the TTFT low regardless of the context length, which reduces TTFT by up to 73% compared to a context miss. It also shows that the TTFT of a hit barely increases when the new prompt length is smaller than 500, emphasizing the advantage of a context hit. These result indicate that a 7 percent hit ratio increase can translate to 7% of requests seeing their TTFT reduced by 42–75% in the emulated environment.

### 4.6 Memory and Computational Overhead

We reserved 1 GB of GPU memory for LPC's predictor and deduct 1 GB from LPC's prefix cache as described in Section 4.1. In this case, LPC takes about 1.25% of the GPU memory running our evaluations. A sensitivity study analyzing the impact of using varying memory for LPC is provided in the appendix.

The GPU computational overhead of LPC is minimal due to the high efficiency of the predictor model, which can process up to 1,000 predictions per second per GPU. A detailed instrumentation is also presented in the appendix.

## 5 Related Work

**Prefix Caching and Eviction Policies.** Many works have explored optimizing LLM inference with prefix caching, but none have placed an emphasis on how eviction policies adjust to the LLM workloads. PromptCache [Gim et al., 2024] reuses common prompt segments with precomputed attention states. RAGCache [Jin et al., 2024] proposes prefix-aware Greedy Dual-Size Frequency eviction policy for RAG but not for conversational workload. CacheBlend [Yao et al., 2025] uses prefix cache and cross SGLang [Zheng et al., 2024] builds RadixAttention to cache KV sequences in a radixtree for efficient prefix matching, while vLLM [Kwon et al., 2023] maps logical KV blocks to a global physical cache. Both adopt a least-recently used (LRU) eviction strategy. Dynamo [NVIDIA Research Team, 2025] uses first-in-first-out (FIFO) eviction, while CachedAttention [Gao et al., 2024] presents a scheduler-aware eviction scheme based the queue of incoming requests for offline batched inference. Marconi [Pan et al., 2025] factors in computational savings for LRU eviction, but the approach is tailored to state-space models and does not extend to transformer architectures.

**ML-based Caching Algorithms.** Existing caching research explores machine learning as an approximate to optimal eviction decisions. LeCaR [Vietri et al., 2018] and CACHEUS [Rodriguez et al., 2021] apply online learning and expert-ensemble frameworks to further learn cache eviction policies. LRB [Song et al., 2020] , MAT [Yang et al., 2023], and HALP [Song et al., 2023] leveraging machine learning in content delivery networks, demonstrates a 10% to 20% hit ratio improvement and cost saving over the best heuristic algorithm and LRU. However, these algorithms cannot be directly applied to LLM prefix caches due to the large difference in their caching paradigms.

**Learned Optimizations for LLM Inference.** Recent LLM inference systems use learned approaches to improve performance. [Zheng et al., 2023b] adds instructions for predicting prompt length in order to group requests with similar latencies, leading to a throughput increase compared to regular batched inference. Fu et al. [2024] and Qiu et al. [2024a,b] train smaller separate models to predict output length, finding that shortest-job-first batching reduces per-token latency significantly compared to first-come-first-serve. Finally, Jain et al. [2025] explores a heuristic-guided reinforcement learning-based router for workload-aware scheduling. Our work on LPC complement this work by focusing its learned optimizations on a different component of LLM inference: prefix caching.

## 6 Conclusion and Future Work

This paper presents the first learned prefix cache LPC to improve LLM inference efficiency. It uses ML on history data to predict conversation continuation probability to improve the cache hit ratio over LRU based approach.

Our evaluations show that LPC consistently outperforms the baseline in all three workloads, achieving up to 43% lower cache size utilization. The improved high-ratios translate to substantially lower latency and an 11% prefilling throughput increase. This is a step towards intelligent and adaptive memory management in LLM serving systems.

While our approach demonstrates significant improvements over LRU, this study has several limitations that we plan to address in future work. First, our evaluation relies on workloads augmented with synthetic timestamps, which may not fully capture real-world user behavior. Second, our comparison is primarily against LRU; evaluating against a broader range of heuristic eviction strategies would offer a more comprehensive view. Third, while the principles of LPC are model-agnostic, our experiments focused on a single model family; demonstrating its effectiveness across different architectures (e.g., Llama, Mistral) would further strengthen our claims of generality.

This work also opens several promising directions for future research. A finer-grained analysis of hit ratios on specific domains (e.g., coding, agentic tasks) could provide deeper insights into which conversational patterns are most amenable to caching. Another exciting avenue is extending LPC to handle multi-modal inputs. This could be achieved by replacing the separate text encoder with the main LLM's own internal hidden states as a rich, holistic embedding for the predictor, making the framework highly adaptable to the future of multi-modal generative AI. These directions represent promising next steps for advancing learned prefix caching in LLM serving systems.

## Acknowledgments

This material is based upon work supported by the National Science Foundation under Grant Number CNS-2321723. Any opinions, findings, and conclusions or recommendations expressed in this material are those of the author(s) and do not necessarily reflect the views of the National Science Foundation.

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

# A  Additional Experiments

## A.1  LPC Computational Overhead

In this section, we extend Section 4.6 to address the computational overhead of our proposed method LPC.

**Metrics.**  We evaluate GPU computational activity using three fine-grained metrics collected by NVIDIA Data Center GPU Manager: SM Active, SM Occupancy, and Tensor Pipe Active. SM Active indicates the proportion of cycles during which at least one warp was assigned to any Streaming Multiprocessor (SM), reflecting overall SM computational engagement. SM Occupancy measures the ratio of active resident warps to the theoretical maximum warps per SM, averaged across SMs. Tensor Pipe Active represents the ratio of cycles where any tensor pipe was active.

**Setup.**  We run three configurations: our proposed LPC method, the baseline LRU method, and the LRU w/ predictor configuration. The LRU w/ predictor configuration integrates the LPC predictor with the standard LRU policy; this setup is specifically designed to measure the predictor's intrinsic computational overhead while maintaining a cache hit ratio comparable to the baseline LRU. The experiments are conducted on the LMSys dataset with a 40 GB cache size. The chat interval parameter $\lambda_{chat}$ is 100.

Table 2: GPU computational activity on LMSys with 40 GB cache size.

| Metric | LPC | LRU | LRU w/ predictor |
|---|---|---|---|
| SM Active (%) | 39.70 | 41.74 | 42.04 |
| SM Occupancy (%) | 7.63 | 8.05 | 8.12 |
| Tensor Pipe Active (%) | 11.42 | 12.05 | 12.07 |
| Hit Ratio (%) | 46.50 | 40.90 | 40.90 |

**Results.**  The results in Table 2 show that LPC reduces the overall GPU computations by about 5%. Specifically, LPC achieves a 4.89% decrease in SM Active, a 5.22% decrease in SM Occupancy, and a 5.23% decrease in Tensor Pipe Active, underscoring its effectiveness in reducing the overall computational load due to its higher prefix cache hit ratio. Second, to assess the computational overhead introduced by the LPC predictor, we compared the LRU baseline with and without running an LPC predictor. The results indicate that the predictor's overhead is minimal: SM Active increased by only 0.72%, SM Occupancy by 0.87%, and Tensor Pipe Active by a mere 0.17% when the predictor was added to LRU. This confirms that the LPC predictor can be integrated to LPC with a marginal computational overhead.

## A.2  LPC Memory Overhead Sensitivity Analysis

We analyze the sensitivity of LPC's performance to its 1 GB GPU memory overhead by scaling the reserved memory for the predictor by $0\times$, $1\times$, $2\times$, and $4\times$. Figure 8 shows the impact on hit ratio using the same setup as Section 4.3.

The results show that the GPU memory allocated to LPC's predictor has a $0.6\%$ to $8.1\%$ impact on its hit ratio. This is observed by comparing LPC ($0\times$) with LPC ($1\times$). The hit ratio sees a reduction ranging from $0.6\%$ to $2.4\%$, at a cache size of $16 \times 10^4$ tokens (40 GB for H100 after loading Qwen3-32B-FP8). The hit ratio impact is more pronounced at smaller cache capacities, as the reduction in hit ratio when moving from LPC ($0\times$) to LPC ($1\times$) ranges from $2.3\%$ to $8.1\%$, at a cache size of $6.4 \times 10^4$ tokens (16 GB). These findings indicate that though LPC's prediction mechanism is fundamentally superior to LRU, minimizing its direct GPU memory overhead is crucial for maximizing its cache hit ratio benefits, especially when GPU memory resources are limited.

Despite the internal overhead variations, LPC consistently outperforms LRU. When varying LPC's predictor overhead from $0\times$ to $4\times$, LPC achieves $4.3\%$ to $16.7\%$ higher hit ratios than LRU at the cache size of $16 \times 10^4$ tokens (40 GB). At the smaller cache size of $6.4 \times 10^4$ tokens (16 GB), LPC achieves $5.8\%$ to $119.0\%$ higher hit ratios than LRU. These findings indicate that LPC's prediction mechanism is fundamentally superior to LRU.

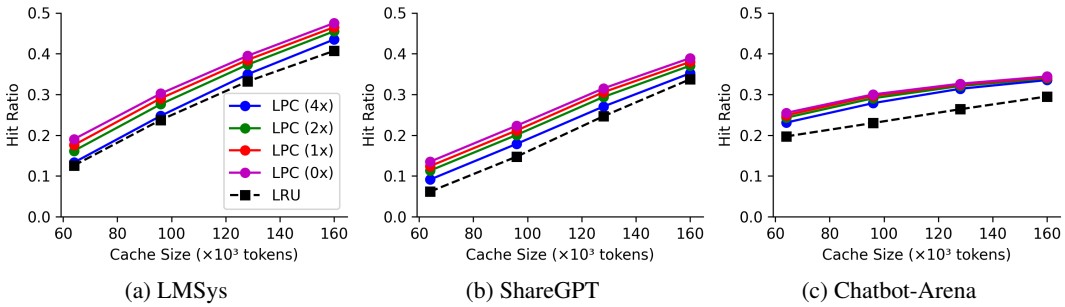

(a) LMSys  (b) ShareGPT  (c) Chatbot-Arena

Figure 8: Sensitivity of LPC's cache hit ratio to its internal GPU memory overhead, compared with LRU. LPC ($N\times$) denotes LPC configured with $N$ times of its actual GPU memory requirement.

## A.3 LPC Predictor Accuracy and Ablation Study

We evaluate the effectiveness of the LPC predictor using the Matthews Correlation Coefficient (MCC) and macro-averaged F1 score (F1 macro). MCC is a robust metric for binary classification with imbalanced classes. The F1 macro computes the F1 score independently for each class and then averages them. LPC$_{\text{turns}}$ is an ablation study using only the number of turns as input, where predictions are based on the majority label of training data binned by turn count.

Table 3: Prediction correctness of LPC predictor and an ablation study using only the number of turns as the input (LPC$_{\text{turns}}$). Higher values are better for all metrics.

| Dataset | Method | MCC | F1 macro |
|---|---|---|---|
| LMSys | LPC$_{\text{turns}}$ | 0.3510 | 0.6755 |
| | LPC | 0.**3863** | 0.**6923** |
| ShareGPT | LPC$_{\text{turns}}$ | -0.0012 | 0.4165 |
| | LPC | 0.**2815** | 0.**6407** |
| Chatbot-Arena | LPC$_{\text{turns}}$ | 0.0775 | 0.4741 |
| | LPC | 0.**3689** | 0.**6837** |

The results in Table 3 indicate that the full LPC predictor achieves 0.28 to 0.36 MCC and is always superior to the LPC$_{\text{turns}}$ ablation. This suggests that leveraging the semantic content of the conversation provides valuable signals for predicting continuation likelihood. For instance, on the Chatbot-Arena dataset, LPC shows a substantial improvement in MCC (0.3689 vs. 0.0775) and F1 macro (0.6837 vs. 0.4741) over LPC$_{\text{turns}}$. The negative MCC for LPC$_{\text{turns}}$ on ShareGPT indicates that using only turn count as a predictor is no better than random guessing for that specific dataset. The LMSys dataset shows comparable MCC and F1 macro between LPC and LPC$_{\text{turns}}$ and indicates that a light-weight predictor such as LPC$_{\text{turns}}$ may be strong enough in some scenarios. Overall, the results show the effectiveness of our content-aware features in predicting the conversation continuation.

## B Additional Ablation Studies

### B.1 Ablation on Input Features: User Prompts vs. Model Responses

Our initial design choice to use only user prompts as input was motivated by keeping the predictor lightweight. As shown in Table 4, including the model response provides no significant, consistent improvement and can even slightly degrade performance. We hypothesize this is because the compact embedding model, chosen for efficiency, may lack the capacity to distill useful signals from the often longer and more varied model responses, potentially introducing noise. The user's prompt appears to be a more direct and potent signal for their immediate intent.

Table 4: Ablation study on predictor input features: prompts only vs. prompts plus model responses. Performance is measured by Matthews Correlation Coefficient (MCC) and F1 macro score.

| Dataset | Features | MCC | F1 macro |
|---|---|---|---|
| Chatbot Arena | Prompts + Model Response | 0.3370 | 0.6685 |
| Chatbot Arena | Prompts Only | **0.3485** | **0.6738** |
| LMSys | Prompts + Model Response | 0.3843 | 0.6849 |
| LMSys | Prompts Only | 0.3797 | **0.6897** |
| ShareGPT | Prompts + Model Response | 0.2779 | 0.6286 |
| ShareGPT | Prompts Only | 0.2777 | **0.6348** |

## B.2  Ablation on Prompt Window Length

We chose a default history window of 5 prompts (the current prompt + 4 previous). This was based on our analysis that most conversations in the datasets are short (e.g., 94-99% have five turns or fewer). To validate this choice, we experimented by increasing the prompt window length to 10. Table 5 shows that this did not yield an improvement and often resulted in slightly lower performance. This reinforces that the most recent conversational history is the most predictive for these workloads. This hyperparameter can be tuned for workloads with substantially longer conversations.

Table 5: Ablation study on the prompt window length for the predictor.

| Dataset | Prompt window length | MCC | F1 Macro |
|---|---|---|---|
| Chatbot Arena | 5 | **0.3485** | **0.6738** |
| Chatbot Arena | 10 | 0.3416 | 0.6707 |
| LMSYS | 5 | **0.3797** | **0.6897** |
| LMSYS | 10 | 0.3727 | 0.6862 |
| ShareGPT | 5 | **0.2777** | **0.6348** |
| ShareGPT | 10 | 0.2496 | 0.6167 |

## B.3  Generalizability: Unified vs. Specialized Predictor

A crucial question for practical deployment is whether a single, general-purpose predictor can perform as well as predictors specialized for different data distributions. We tested this by training a "Unified" predictor on a combined dataset and comparing its performance against our default "Specialized" models on each test set. The results in Table 6 show that the unified model is competitive on the Chatbot Arena and LMSys datasets. However, its performance is notably lower on ShareGPT, suggesting that while core conversational patterns are generalizable, certain datasets like ShareGPT may contain unique characteristics that benefit from domain-specific training. A larger model might improve generalizability, but at the cost of higher overhead.

Table 6: Performance comparison of a Unified predictor vs. Specialized predictors.

| Test Dataset | Model Type | MCC | F1 macro |
|---|---|---|---|
| Chatbot Arena | Unified | 0.3357 | 0.6678 |
| Chatbot Arena | Specialized | **0.3485** | **0.6738** |
| LMSys | Unified | **0.3802** | 0.6871 |
| LMSys | Specialized | 0.3797 | **0.6897** |
| ShareGPT | Unified | 0.1271 | 0.5281 |
| ShareGPT | Specialized | **0.2777** | **0.6348** |

## B.4 Hit Ratio Comparisons with Oracle

To assess the upper bound for further potential improvements, we compare the hit ratio of LPC against an Oracle policy, which possesses perfect future knowledge of whether a conversation will continue. Figure 9 presents this comparison alongside LRU across the three datasets for various cache sizes. The evaluation setup is the same as in Section 4.3.

The results indicate that LPC consistently reduces the hit ratio gap between LRU and Oracle. On the LMSys dataset (Figure 9a), the reduction in the gap is from 22% to 30%. On the ShareGPT dataset (Figure 9b), the reduction is from 28% to 64%. The remaining gap to Oracle is only about 0.5 times the current LPC-over-LRU gain at the smallest cache size. We hypothesize that it is because LPC has higher accuracy in high-confidence predictions, and the small cache size is only enough to hold those high-confidence prefixes. For the Chatbot-Arena dataset (Figure 9c), the reduction of the hit ratio gap from LRU to Oracle achieved by LPC ranges from 25% to 28%.

These observations highlight the substantial advancements LPC delivers in prefix caching. While a performance ceiling defined by the Oracle benchmark remains, we contend that LPC effectively captures the vast majority of predictive information available within the conversation text itself. Therefore, we believe that future efforts to substantially close the remaining gap and approach Oracle-level performance will likely need to incorporate new data dimensions, such as user access patterns, moving beyond sole reliance on textual content for prediction.

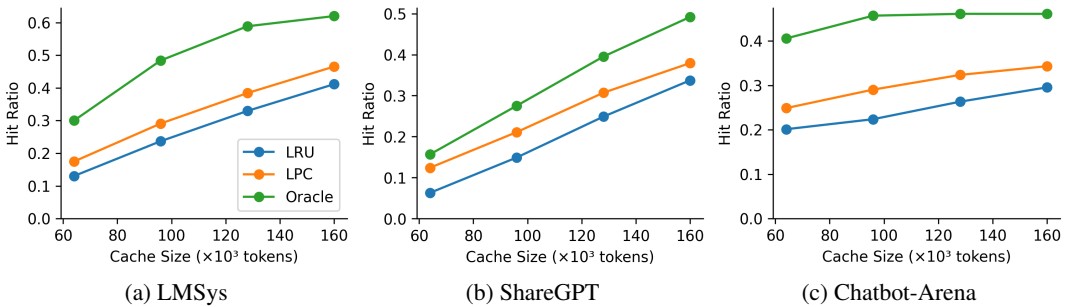

| (a) LMSys | (b) ShareGPT | (c) Chatbot-Arena |

Figure 9: Hit ratio vs. cache size. LPC reduces the gap from LRU to Oracle by 22% to 66%.

## B.5 Experiment Statistical Significance

To assess the statistical significance and stability of our results, we repeated the experimental runs for at least 5 times for each data point. Figure 10 presents the hit ratios with error bars indicating the observed minimum and maximum values across these runs. The analysis of this data reveals that the variations are consistently small. Specifically, the maximum absolute difference between the highest and lowest observed hit ratio for any given configuration (algorithm, dataset, and cache size) is less than 0.013. The relative variation (max-min difference divided by the mean hit ratio) is also small, always below 4.6% and in most cases lower than 1%. This low variance across repeated experiments indicates that the performance differences observed between our proposed method and the baseline are robust and not attributable to random fluctuations, underscoring the statistical significance of our findings.

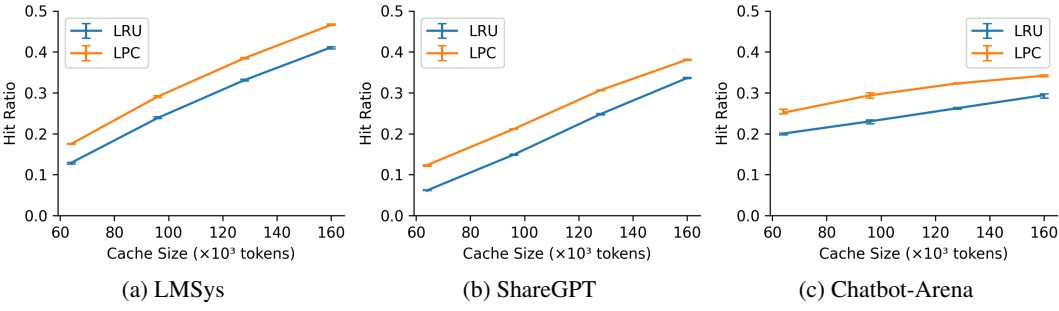

| (a) LMSys | (b) ShareGPT | (c) Chatbot-Arena |

Figure 10: Hit ratio vs. cache size with error bars. The variation in the hit ratio is smaller than 0.013.

