# OpenReview forum: "Learned Prefix Caching for Efficient LLM Inference"
_NeurIPS.cc/2025/Conference — NeurIPS 2025 poster_

### Official Review · Reviewer_hPyr · 2025-06-09

**Clarity:** 3
**Significance:** 3
**Originality:** 2
**Rating:** 4
**Confidence:** 4

**Summary:**

The paper introduces LPC (Learned Prefix Caching), the first learned eviction algorithm for prefix caches in Large Language Model (LLM) inference systems. Traditional prefix caching uses an LRU (Least Recently Used) policy, which underperforms due to its inability to anticipate future usage based on semantic context. LPC overcomes this by predicting the likelihood that a conversation will continue using a lightweight neural model that analyzes recent user prompts. These predictions guide the eviction policy to retain prefixes more likely to be reused, improving cache hit rates. Evaluated on real-world datasets (LMSys, ShareGPT, Chatbot-Arena), LPC achieves 18–47% reductions in required cache size for the same hit ratio compared to LRU, and up to 11% higher prefilling throughput in an emulated environment.

**Questions:**

1. In Line 129, the authors say that “we do not include model responses because they contain less new information than user prompts”. Could you further explain why the model response will not help judge the probability of a follow-up question? For example, if the model’s response is not complete enough, isn’t that the probability that the user will have a follow-up can be higher? Do you have any analysis or ablation on this?

2. In Line 151, the authors say that “the request rate is typically lower than 10 per second per GPU”. Could you explain how did you get the 10/sec number?

3. In Line 160, the authors say that they train “a separate model for each dataset”. Could you explain why do you need separate models for different datasets? Can you train a single predictor for all settings?

**Ethical Concerns:**

["NO or VERY MINOR ethics concerns only"]

**Final Justification:**

The authors response has addressed most of my questions. I will increase my score to 4.

**Limitations:**

yes

**Quality:**

3

**Strengths And Weaknesses:**

[Strengths]
1. The paper proposes a learned eviction strategy tailored to prefix caching in LLM serving – LPC, identifying an important issue in efficient LLM serving while alleviating the limitations of existing methods like LRU.
2. The paper presents adequate analysis on the performance improvements and overheads introduced by the proposed method, especially with the experiments in the appendix.
3. The presentation and visual components of this paper are clear and easy to understand. The authors also include detailed training setups in this paper.

[Weaknesses]
1. Ablations on several important designs are missing. See questions section for more details.
2. The cache hit ratio improvement over the LRU baseline seems to be not very significant, and the overall throughput improvement is a bit marginal.
3. Experiments are limited to specific settings such as QWen-32B. Experiments on different model series and different model sizes may better prove the effectiveness of the design.

---

> ### Author Rebuttal · Authors · 2025-07-31
>
> We sincerely thank the reviewer for their positive assessment and constructive feedback. We are encouraged that you recognized the importance of the problem, the adequacy of our analysis, and the clarity of the paper's presentation. The insightful questions and comments provided are invaluable and will help us to substantially improve the clarity and completeness of our paper.
>
> We address the specific points raised below.
>
> ***
>
> ### **Response to Weaknesses**
>
> #### **1. On the Significance of Performance Improvements**
>
> We thank the reviewer for this critical point. We realize we could have better contextualized the impact of our results. In large-scale LLM serving, seemingly small percentage gains in core metrics lead to substantial downstream effects on performance and cost.
>
> Our method LPC provides a 11% increase in prefilling throughput. In a production environment serving millions of requests, this means serving thousands of additional users per day on the same infrastructure, which can translate to millions of dollars in saved hardware costs annually.
>
> Furthermore, our approach is orthogonal to most other inference optimization techniques. As a cache eviction policy, LPC operates at the system management layer and can be seamlessly combined with architectural improvements like FlashAttention or model-level optimizations like quantization and speculative decoding. The benefits from LPC are therefore additive to gains from these other methods. We will revise the paper to better highlight the practical significance of our results.
>
> #### **2. On the Generality of Experimental Settings**
>
> This is an excellent point. Our current experiments are focused on the Qwen-32B model. We chose it as a representative example of a modern, high-performance LLM. The fundamental principles of LPC are model-agnostic. The predictor learns patterns of *conversation continuation* only from user prompts, not specific to one model's architecture. The model output length is specified by the benchmark dataset. The model output content is not used by the baseline or our algorithm.
>
> However, we agree that demonstrating this effectiveness across different model families (e.g., Llama, Mistral) and sizes would greatly strengthen the paper. We will add comprehensive results in our paper.
>
> #### **3. On Missing Ablations**
>
> Thank you for noting this. We provide detailed answers to your questions below, which include new empirical results for the most important design ablations: the choice to exclude model responses and the use of separate vs. unified predictors.
>
> ***
>
> ### **Responses to Questions**
>
> #### **1. Could you further explain why the model response will not help? Do you have any analysis or ablation on this?**
>
> This is an excellent question. Our initial design choice to exclude model responses was motivated by keeping the predictor as lightweight as possible. To validate this empirically, we conducted an ablation study where we trained a predictor that includes the most recent model response as an additional feature.
>
> **Table 1:** Prompts only vs. Prompts + Model Respons as input features.
>
> | Dataset | Feature | MCC | F1 macro |
> | :--- | :--- | :--- | :--- |
> | Chatbot Arena | Prompts + Model Response | 0.3370 | 0.6685 |
> | Chatbot Arena | Prompts Only | 0.3485 | 0.6738 |
> | LMSys | Prompts + Model Response | 0.3843 | 0.6849 |
> | LMSys | Prompts Only | 0.3797 | 0.6897 |
> | ShareGPT | Prompts + Model Response | 0.2779 | 0.6286 |
> | ShareGPT | Prompts Only | 0.2777 | 0.6348 |
>
> As shown in **Table 1**, including the model response provides no significant, consistent improvement and can even degrade performance (e.g., on Chatbot Arena). We hypothesize this is because the compact embedding model, chosen for efficiency, may struggle to distill useful signals from the longer and more varied model responses. The user's prompt remains the most direct and potent signal of their immediate intent. We will add this table and analysis to the appendix.
>
> #### **2. Could you explain how did you get the 10/sec number?**
>
> Thank you for the question. This figure is based on performance benchmarks of modern LLM serving systems. For example, the **DistServe paper (Zhong et al., OSDI '24)** shows in their Figure 1 that for a 13B model on an A100 80GB GPU with an average input length of 512 tokens, the prefilling throughput is less than 10 req/s. Since many production models are significantly larger than 13B in the prefilling phase, this is a reasonable, and often conservative, estimate for workloads with moderate prompt lengths. We will add this citation and clarification to the paper.
>
> #### **3. Could you explain why you need separate models for different datasets? Can you train a single predictor for all settings?**
>
> This is a crucial question about the practical deployment of LPC. To test the data sensitivity and generalizability of our method, we conducted a key experiment where we trained a single "unified predictor" on a combined dataset and compared its performance against the specialized models.
>
> **Table 2:** Unified predictor trained on a combined dataset vs. Specialized models.
>
> | Test Dataset | Model Type | MCC | F1 macro |
> | :--- | :--- | :--- | :--- |
> | Chatbot Arena | Unified | 0.3357 | 0.6678 |
> | Chatbot Arena | Specialized | 0.3485 | 0.6738 |
> | LMSys | Unified | 0.3802 | 0.6871 |
> | LMSys | Specialized | 0.3797 | 0.6897 |
> | ShareGPT | Unified | 0.1271 | 0.5281 |
> | ShareGPT | Specialized | 0.2777 | 0.6348 |
>
> The results in **Table 2** show that the unified model's performance is competitive with specialized models on the **Chatbot Arena** and **LMSys** datasets. However, on the **ShareGPT** dataset, the unified model's performance is notably lower. This suggests that while core conversational patterns are generalizable, certain datasets like ShareGPT may contain unique characteristics that benefit from domain-specific training. It is also possible that each dataset has unique request patterns that contradicts other datasets, leading to performance drop if it is trained in a mixed way. We hypothesize that a larger model or a full fine tuning of the embedding model can increase the learning capacity and make it have less performance drop on ShareGPT. However, the expensive cost prevent us from adopting the above method. Therefore, we still need a separate model for the best accuracy and overhead trade-off. We will add this important result and discussion to the paper.
>
> ***
>
> Once again, we thank the reviewer for their constructive feedback. We believe that by incorporating these clarifications, the final paper will be substantially stronger. We hope our responses have addressed the reviewer's concerns and have reinforced the value and technical soundness of our contribution.

---

> > ### Comment · Reviewer_hPyr · 2025-08-04
> >
> > Thank you for the response! I have increased my score to 4.

---

### Official Review · Reviewer_X9iT · 2025-06-28

**Clarity:** 3
**Significance:** 2
**Originality:** 2
**Rating:** 4
**Confidence:** 3

**Summary:**

The paper introduces Learned Prefix Caching (LPC), a new method for managing the cache used during Large Language Model (LLM) inference. The paper notes that the commonly used least-recently-used (LRU) algorithm is not ideal for this purpose, as it does not account for the conversational context. LPC uses a learned predictor to analyze conversational content and determine which conversations are likely to continue, which in turn helps decide which data to keep in the cache. The evaluations show that LPC consistently outperforms LRU, leading to a significant reduction in the required cache size for the same performance and an improvement in prefilling throughput.

**Questions:**

1. Besides the evaluation on cache hit ratio and throughput, conducting an evaluation of the classifier accuracy would provide a more comprehensive understanding of the predictor's performance. I am also curious why you only train the MLP part of the predictor? Full fine-tuning would likely yield better results without affecting the deployment overhead.

2. A growing workload of LLM serving systems is to handle multi-modal inputs, such as images and videos. How does the proposed method handle multi-modal inputs? Is it possible to extend the LPC predictor to work with multi-modal data?

**Ethical Concerns:**

["NO or VERY MINOR ethics concerns only"]

**Final Justification:**

This paper tackles an important problem as LLM inference traffic is growing rapidly; therefore, I increase my score to 4. However, the technical part does not endow sufficient novelty that prevents me from assigning higher scores.

**Limitations:**

yes

**Quality:**

2

**Strengths And Weaknesses:**

### Strengths

1. This paper studies a meaningful and practical problem in LLM serving systems, specifically the inefficiency of the LRU algorithm for prefix caching. As the inference demand for LLMs grows, optimizing cache management is crucial for reducing latency and compute costs.

2. Extensive evaluations are conducted across three real-world datasets (although the timestamps are synthetic), demonstrating the effectiveness of the proposed method. The results show that LPC can reduce the required cache size by 18-47% for the same hit ratio and improve LLM prefilling throughput by 11%. The paper also assesses the minimal overhead of the LPC predictor, showing that it is a practical solution.

3. The paper is generally well-written and easy to follow.

### Weaknesses

1. The LPC predictor introduces additional complexity and overhead to the LM serving system. While the paper claims to observe substantial performance improvements on several datasets, there always exists a worst-case scenario where no shared prefix is found and all the computational overhead of the predictor is wasted. The LRU strategy, while sub-optimal, is much simpler and does not require any training.

2. Although the problem setting is well-motivated and novel, the proposed solution itself is not particularly innovative. The idea of training a classifier over pre-trained embeddings has been explored in various contexts. The novelty of this work lies more in its application to LLM prefix caching rather than in the underlying methodology.

---

> ### Author Rebuttal · Authors · 2025-07-31
>
> We thank the reviewer for their detailed feedback and for recognizing that our work addresses a meaningful and practical problem with extensive evaluations. The insightful questions and comments provided are invaluable and will help us to substantially improve the clarity and completeness of our paper.
>
> We address the specific points raised below.
>
> ***
>
> ### **Response to Weaknesses**
>
> #### **1. On Complexity and Worst-Case Overhead**
>
> We agree with the reviewer that LRU is simpler. However, we argue this trade-off between simplicity and performance is highly favorable for LPC in its target domain.
>
> Our work focuses on **multi-turn conversational workloads**, where reusing previous turns as a prefix is inherent to the application. The "worst-case" scenario of no prefix reuse is therefore rare in this setting. This is validated by recent workload characterizations; for example, a study of two production traces from a large cloud provider, detailed in **"KVCache Cache in the Wild: Characterizing and Optimizing KVCache Cache at a Large Cloud Provider" (Li et al., arXiv '25)**, found that prefix reuse is common, with ideal hit rates of **62%** and **54%**. This confirms our problem setting is highly practical.
>
> Furthermore, there is a clear industry trend of replacing simple heuristics like LRU with more sophisticated, learned policies when the cost savings outweigh the complexity. A prominent example is the deployment of a machine learning-based caching policy in **YouTube's content delivery network, as described in "HALP: Heuristic Aided Learned Preference Eviction Policy for YouTube Content Delivery Network" (Song et al., NSDI '23)**. Similarly, LPC's minimal overhead is justified by its significant gains in memory efficiency and throughput.
>
> #### **2. On Novelty**
>
> We appreciate the reviewer's perspective on novelty. We agree that the core machine learning components, such as using classifiers on pre-trained embeddings, are established techniques.
>
> The primary novelty of our work lies in:
> 1.  **Problem Formulation:** Identifying and formalizing the specific inefficiency of LRU for the stateful, conversational nature of LLM prefix caching.
> 2.  **System Design:** Designing a lightweight and practical *end-to-end system* (LPC) that effectively applies a predictive approach to this new and important domain.
> 3.  **Demonstrated Impact:** Proving that this application leads to significant, measurable improvements in key system metrics.
>
> Therefore, the contribution is in the novel application, problem formulation, and effective system design, rather than the invention of a new ML algorithm from scratch.
>
> ***
>
> ### **Responses to Questions**
>
> #### **1.1 On Predictor Accuracy**
>
> We have evaluated the predictor's performance using the F1 score and Matthews Correlation Coefficient (MCC) on line 511 in our paper's Appendix, which confirm that our predictor using user prompt as the input is effective and necessary. We paste the results here for your convinience.
>
> | Dataset | Method | MCC | F1 macro |
> | :--- | :--- | ---: | ---: |
> | LMSys | LPC with turns input only | 0.3510 | 0.6755 |
> | LMSys | LPC | **0.3863** | **0.6923** |
> | ShareGPT | LPC with turns input only | -0.0012 | 0.4165 |
> | ShareGPT | LPC | **0.2815** | **0.6407** |
> | Chatbot-Arena | LPC with turns input only | 0.0775 | 0.4741 |
> | Chatbot-Arena | LPC | **0.3689** | **0.6837** |
>
> #### **1.2 MLP-only Training**
>
> Thank you for the excellent question. We chose to train only the MLP head to ensure the entire framework remains lightweight and practical for deployment. In a production environment, the predictor would need to be periodically retrained (e.g., daily) on recent query logs to adapt to evolving usage patterns.
>
> Full fine-tuning of the entire embedding model on a large daily corpus could take several hours, making frequent retraining operationally expensive. In contrast, training only the tiny MLP head is extremely fast—typically taking only ~10 minutes. This efficiency makes daily online retraining feasible, ensuring LPC remains adaptive to user behavior without incurring significant computational cost.
>
> #### **2. On Handling Multi-modal Inputs**
>
> This is a fantastic question and we are working on it with a new learning paradigm:
>
> This new learning paradigm of LPC replaces the text embedding model with **the main LLM's own internal state**. To be specific, after the LLM processes a prompt, we extract the final layer's hidden states (e.g., from the KV cache) to use as a rich embedding for our predictor.
>
> This improved method can be directly applied to multi-modal input because the LLM's internal state is a holistic representation of all inputs it has processed, whether they are text, images, or other data types. This eliminates the need for a separate encoder, reducing system complexity and overhead while providing a powerful, context-aware signal to the LPC predictor. This makes the LPC framework highly adaptable to the future of multi-modal generative AI. We will add this to our future work section.
>
> ***
>
> Once again, we thank the reviewer for their constructive feedback. We believe that by incorporating these clarifications, the final paper will be substantially stronger. We hope our responses have addressed the reviewer's concerns and have reinforced the value and technical soundness of our contribution.

---

> > ### Comment · Reviewer_X9iT · 2025-08-04
> >
> > Thanks for the response! This paper tackles an important problem as LLM inference traffic is growing rapidly; therefore, I increase my score to 4. However, the technical part does not endow sufficient novelty that prevents me from assigning higher scores.

---

### Official Review · Reviewer_z2XZ · 2025-07-03

**Clarity:** 2
**Significance:** 3
**Originality:** 3
**Rating:** 4
**Confidence:** 3

**Summary:**

This paper introduces a learned prefix cache framework that analyzes conversation content and evicts outdated cache information smartly. Different from LRU that evicts the least recent cache information, the method utilizes a lightweight ML predictor to analyze the historical user prompts to estimate the possibility of the conversition continuing. The framework then leverages the possibiltiy to guide eviction decisions in the prefix cache.

**Questions:**

1. How do you determine the value of scale?

2. Can you provide more explanations about the probability decay function?

3. Can you compare LPC with LRU on throughput?

**Ethical Concerns:**

["NO or VERY MINOR ethics concerns only"]

**Limitations:**

Yes

**Quality:**

3

**Strengths And Weaknesses:**

Strengths:

1. The method adapts to online scenarios and can handle high query volumes, making it practical for production LLM serving systems where low latency and high throughput are critical.

2. The authors introduce a novel eviction algorithm that analyzes the historical prompts to predict conversation continuation.

3. The method achieves comparable hit ratios compared to LRU but reduces the cache size by 18–47%.

Weakness:


1. The method relies on the last 4 user prompts for prediction, which might fail to capture the complex interactions between users and the chatbot. Users might formulate new questions based on the LLM's responses rather than just their previous prompts. Thus, it might be biased to only use the 4 latest prompts to estimate if the chat is finished or not.

2. The 4 prompt window might miss long-term conversational patterns. In a case when users seek thorough explanations from a chatbot, usually it is difficult to determine if the users' questions are fully addressed with 4 questions.


3. The approach trains separate predictor models for each dataset (Line 161), which poses practical challenges in real-world LLM serving. Real-world LLM serving systems handle highly diverse queries across users and topics, making dataset-specific training almost impractical for production deployment. Is your proposed method data-sensitive? Is it possible to handle a more general case?

---

> ### Author Rebuttal · Authors · 2025-07-31
>
> We are sincerely grateful to the reviewer for their thoughtful and positive assessment of our work. We are particularly encouraged by the recognition of LPC's practicality for production systems and its significant 18%-47% cache size reduction over LRU. The insightful questions and comments provided are invaluable and will help us to substantially improve the clarity and completeness of our paper.
>
> We address the specific points raised below.
>
> ***
>
> ### **Response to Weaknesses**
>
> #### **1.1 Ablation Study on User Prompts vs. Model Responses**
>
> We thank the reviewer for this excellent and very important point. We agree that the choice of input features is critical, and we appreciate the suggestion that model responses could potentially provide useful signals for predicting user follow-up. Our initial design choice to use only user prompts was motivated by keeping the predictor as lightweight as possible.
>
> To investigate this empirically, we conducted a new ablation study during the rebuttal period. In this study, we trained a predictor that includes the most recent model response as an additional feature. The results are summarized in **Table 1**.
>
> **Table 1:** Prompt only vs. Prompt + model responses as input features. The predictor performance is measured by The F1 score and Matthews Correlation Coefficient (MCC)
>
> | Dataset | Features | MCC | F1 macro |
> | :--- | :--- | :--- | :--- |
> | Chatbot Arena | Prompts + Model Response | 0.3370 | 0.6685 |
> | Chatbot Arena | Prompts Only | 0.3485 | 0.6738 |
> | LMSys | Prompts + Model Response | 0.3843 | 0.6849 |
> | LMSys | Prompts Only | 0.3797 | 0.6897 |
> | ShareGPT | Prompts + Model Response | 0.2779 | 0.6286 |
> | ShareGPT | Prompts Only | 0.2777 | 0.6348 |
>
> The results show that including the model response provides no significant, consistent improvement. We hypothesize this is because the compact embedding model (`multilingual-e5-small`), chosen for its efficiency, may lack the capacity to effectively distill useful signals from the often longer and more varied model responses, potentially introducing noise. The user's prompt appears to be a more direct and potent signal. We will add this result and discussion to the paper.
>
> #### **1.2 Ablation Study on the Prompt Window Length**
>
> We thank the reviewer for raising the excellent question about the 4-prompt window. Our analysis of the 3 public datasets shows that the vast majority of conversations are short. For instance, in the Chatbot Arena and LMSys datasets, 99.6% and 94% of conversations have five turns or fewer, respectively. To further validate this choice, we experimented by increasing the prompt window length from 5 (our default) to 10. As shown in **Table 2**, this change did not yield an improvement and often resulted in slightly lower performance. This reinforces that the most recent history is the most predictive for the available workload. Nevertheless, this hyperparameter can be mannualy tuned if LPC is applied on a workload with substentially more turns per conversation. We will add this result and discussion to the paper.
>
> **Table 2:** Prompt window length.
> | Dataset | Prompt window length | MCC | F1 Macro |
> | :--- | :--- | :--- | :--- |
> | Chatbot Arena | 5 | 0.3485 | 0.6738 |
> | Chatbot Arena | 10 | 0.3416 | 0.6707 |
> | LMSYS | 5 | 0.3797 | 0.6897 |
> | LMSYS | 10 | 0.3727 | 0.6862 |
> | ShareGPT | 5 | 0.2777 | 0.6348 |
> | ShareGPT | 10 | 0.2496 | 0.6167 |
>
> #### **2. On the Practicality and Generalizability of the Predictor**
>
> We thank the reviewer for raising this crucial question regarding the practical deployment and data sensitivity of LPC. We agree that a generalizable model is essential for production systems.
>
> To directly test this, we conducted a key experiment where we trained a single "unified predictor" on a combined dataset and compared its performance against models trained on specialized datasets (our default).
>
> **Table 3:** Unified predictor trained on a combined dataset vs. Specialized predictor.
>
> | Test Dataset | Model Type | MCC | F1 macro |
> | :--- | :--- | :--- | :--- |
> | Chatbot Arena | Unified | 0.3357 | 0.6678 |
> | Chatbot Arena | Specialized | 0.3485 | 0.6738 |
> | LMSys | Unified | 0.3802 | 0.6871 |
> | LMSys | Specialized | 0.3797 | 0.6897 |
> | ShareGPT | Unified | 0.1271 | 0.5281 |
> | ShareGPT | Specialized | 0.2777 | 0.6348 |
>
> The results in **Table 3** show that the unified model's performance is competitive with specialized models on the **Chatbot Arena** and **LMSys** datasets. However, on the **ShareGPT** dataset, the unified model's performance is notably lower. This suggests that while core conversational patterns are generalizable, certain datasets like ShareGPT may contain unique characteristics that benefit from domain-specific training. It is also possible that each dataset has unique request patterns that contradicts other datasets, leading to performance drop if it is trained in a mixed way. We hypothesize that a larger model or a full fine tuning of the embedding model can increase the learning capacity and make it have less performance drop on ShareGPT. However, the expensive cost prevent us from adopting the above method. Therefore, we still need a separate model for the best accuracy and overhead trade-off. We will add this important result and discussion to the paper.
>
> ***
>
> ### **Responses to Questions**
>
> #### **1. How do you determine the value of `scale`?**
>
> Thank you for this insightful question. The `scale` hyperparameter was tuned empirically on our validation sets. We found that a good rule of thumb is to set $s$ to be the inverse of the average turn interval within a conversation. Since this interval is approximately 100 seconds across our datasets, we set the scale hyperparameter $s = 10^{-2}$ in all our experiments. We will add a sentence to Section 3.4.2 of the paper to make this tuning process and rationale more explicit.
>
> #### **2. Can you provide more explanations about the probability decay function?**
>
> Thank you for this question. The probability decay function is designed to complement our ML predictor with a temporal component. Its purpose is to ensure that a prefix whose continuation probability might have been overestimated by the predictor does not remain in the cache indefinitely if it goes unused.
>
> The function adjusts the initial predicted probability ($p_{\text{orig}}$) based on an exponential decay factor, $d$. The current probability ($p_{\text{cur}}$) is calculated as:
> $$p_{\text{cur}} = \frac{p_{\text{orig}} \cdot d}{p_{\text{orig}} \cdot d + (1 - p_{\text{orig}})}$$The decay factor $d$ decreases as the time since the prefix was last accessed ($\Delta t = t_{\text{cur}} - t_{\text{last}}$) increases:$$d = \exp(-\Delta t \cdot s)$$
> As $\Delta t$ increases, $d$ approaches zero, which smoothly reduces the prefix's score ($p_{\text{cur}}$) and makes it a more likely candidate for eviction.
>
> #### **3. Can you compare LPC with LRU on throughput?**
>
> Yes, the reviewer is correct to ask for this comparison. We provide this analysis in Section 4.5 and Figure 6 of the main paper, where our microbenchmark evaluates prefilling throughput as a direct function of the cache hit ratio.
>
> Specifically, Lines 293-295 and Figure 6 show that a hit ratio improvement from 35% to 42%—a gain that LPC consistently achieves over LRU in our tests (e.g., Figure 5a)—directly translates to a throughput increase from 10.01 req/s to 11.11 req/s. This demonstrates an 11% higher prefilling throughput for LPC over LRU in that environment. We will revise the text in Section 4.5 to make this comparison even more direct and explicit.
>
> ***
>
> Once again, we thank the reviewer for their constructive feedback. We believe that by incorporating these clarifications, the final paper will be substantially stronger. We hope our responses have addressed the reviewer's concerns and have reinforced the value and technical soundness of our contribution.

---

> > ### Comment · Reviewer_z2XZ · 2025-08-06
> >
> > Thank you for your explanation. I will keep my score.

---

### Official Review · Reviewer_fKMe · 2025-07-12

**Clarity:** 3
**Significance:** 3
**Originality:** 3
**Rating:** 4
**Confidence:** 3

**Summary:**

This work propose to replace the LRU cache eviction in LLM serving by a learnable component, to improve the kv cache hit ratio for better efficiency. Experiments show that the proposed Learned Prefix Caching can achieve 18-47% reductions in required cache sizes for equivalent hit ratios and has an 11% improvement in LLM prefiling throughput in an emulated environment.

**Questions:**

NA

**Ethical Concerns:**

["NO or VERY MINOR ethics concerns only"]

**Quality:**

3

**Strengths And Weaknesses:**

Strength:
1) The paper is clear and well written.
2) As the improvement of LLMs, the serving cost is growing fast due to more usage across the world. It is very important to improve the model serving efficiency. In addition to that, the GPU memory is growing slowly but the model size is growing fast. Using memory smarter is good.


Weakness:
1) Since there is an additional round of training before serving LLM, there is risk that model is easier to overfit and the improvement might be way smaller on OOD data.
2) As the community is switching to scale RL and test-time compute (more frontier models like Gemini and O3 are using thinking mode by default), more serving cost is from decoding stage instead of prefilling.
3) It would be good to have a finer grained results like the hit ratio on the prompts from different domains (e.g. agent, coding, chatbot...)

---

> ### Author Rebuttal · Authors · 2025-07-31
>
> We are very grateful to the reviewer for their time and for providing such a positive and insightful assessment of our work. We are encouraged that they found the paper "clear and well written" and recognized the importance of improving LLM serving efficiency. Their thoughtful comments have given us an excellent opportunity to clarify the robustness and applicability of our approach.
>
> We address the specific weaknesses raised below.
>
> ***
>
> ### **1. On the Risk of Overfitting and Out-of-Distribution (OOD) Performance**
>
> We agree this is a crucial point for any learned system deployed in a dynamic environment. The risk of performance degradation due to model staleness or out-of-distribution (OOD) data is real, but it is a well-understood challenge in production machine learning that can be effectively managed.
>
> #### **1.1 Mitigation Through Periodic Retraining**
> Our primary strategy for mitigating this risk is **periodic retraining**. On line 172 in our paper, we propose to retrain LPC daily. This is a standard practice in ML-based cache policy to continuously adapt to evolving query patterns and user behaviors. For example, as described in "HALP: Heuristic Aided Learned Preference Eviction Policy for YouTube Content Delivery Network" (Song et al., NSDI '23)**, retraining the ML-based cache policy weekly is sufficient for Youtube. It indicates that the cache workload is relatively stable in their production environement.
>
> #### **1.2 Adaptive Strategies for High-Risk Scenarios**
> For use cases where the OOD risk is particularly high—such as applications driven by rapidly changing current events or highly dynamic agentic workflows—the system can leverage **online learning**. In this paradigm, the model would update itself incrementally with each new data point or mini-batch that arrives. This allows for near-real-time adaptation, ensuring the system remains robust even in the face of rapid and continuous distributional drift.
>
> We will inlcude these discussions in our paper.
>
> ***
>
> ### **2. On the Relevance of Prefill vs. Decoding Stages**
>
> This is an excellent, forward-looking point. We agree with the reviewer that the computational landscape of LLM serving is evolving, with models like Gemini and others employing more complex, multi-step reasoning that increases the relative cost of the decoding stage.
>
> However, we believe that optimizing the prefill stage remains a crucial and highly relevant challenge for two key reasons:
> 1.  **Prefill is Universal and Frequent:** The vast majority of current LLM applications (e.g., interactive chatbots, RAG systems, summarization) are multi-turn and rely heavily on repeated prefilling. Every turn in a conversation after the first requires a new prefill operation. Improving the KV cache hit ratio directly reduces the latency and computational cost of every single one of these turns.
> 2.  **Complementary, Not Competing:** Optimizations for prefilling and decoding are not mutually exclusive; they are complementary. By making the prefill stage 11% faster, as shown in our work, we free up valuable GPU resources that can then be dedicated to handling longer and more expensive decoding steps.
>
> Therefore, while the cost distribution may shift in future models, our work provides a significant and immediate efficiency gain for a fundamental operation that is ubiquitous in today's LLM serving environments.
>
> ***
>
> ### **3. On Finer-Grained, Domain-Specific Results**
>
> This is a valuable suggestion. For this initial work, our goal was to establish the foundational viability of LPC across broad, widely-used conversational datasets. However, a finer-grained analysis of the hit ratio on specific domains (e.g., coding, agentic tasks, creative writing) is an excellent and logical next step for this research. Such an analysis would provide deeper insights into which types of conversational patterns are most amenable to caching. We will add this as a key direction for future work in our conclusion.
>
> ***
>
> Once again, we thank the reviewer for their constructive feedback. We believe that by incorporating these clarifications, the final paper will be substantially stronger. We hope our responses have addressed the reviewer's concerns and have reinforced the value and technical soundness of our contribution.

---

### Decision · Program_Chairs · 2025-09-17

**Decision:**

Accept (poster)

**Comment:**

The paper presents Learned Prefix Caching, a neural eviction policy for LLM prefix caches. Unlike LRU, LPC uses a lightweight predictor over recent user prompts to estimate continuation likelihood. Experiments show LPC reduces cache size needs by 18–47% (to keep the same hit ratios) and improves throughput by 11%. This work is practical and clearly addresses a problem in multi-turn LLM serving. However, its technical novelty is limited since the core method is a standard classifier applied in a new setting.

Reviewers agreed the problem is important, the paper is clear, and the evaluation is solid. They noted, however, that performance gains are modest, ablations were initially incomplete, and evaluation was restricted to specific LLMs. The rebuttal addressed most of these points, adding ablations and clarifying design choices, which slightly improved sentiment. Overall, while the contribution is somewhat incremental, the demonstrated efficiency benefits justify acceptance.